# Conservative Contextual Linear Bandits

**Abbas Kazerouni**
Stanford University
abbask@stanford.edu

**Mohammad Ghavamzadeh**
DeepMind
ghavamza@google.com

**Yasin Abbasi-Yadkori**
Adobe Research
abbasiya@adobe.com

**Benjamin Van Roy**
Stanford University
bvr@stanford.edu

## Abstract

Safety is a desirable property that can immensely increase the applicability of learning algorithms in real-world decision-making problems. It is much easier for a company to deploy an algorithm that is safe, i.e., guaranteed to perform at least as well as a baseline. In this paper, we study the issue of safety in contextual linear bandits that have application in many different fields including personalized recommendation. We formulate a notion of safety for this class of algorithms. We develop a safe contextual linear bandit algorithm, called *conservative linear UCB* (CLUCB), that simultaneously minimizes its regret and satisfies the safety constraint, i.e., maintains its performance above a fixed percentage of the performance of a baseline strategy, uniformly over time. We prove an upper-bound on the regret of CLUCB and show that it can be decomposed into two terms: **1)** an upper-bound for the regret of the standard linear UCB algorithm that grows with the time horizon and **2)** a constant term that accounts for the loss of being conservative in order to satisfy the safety constraint. We empirically show that our algorithm is safe and validate our theoretical analysis.

## 1 Introduction

Many problems in science and engineering can be formulated as decision-making problems under uncertainty. Although many learning algorithms have been developed to find a policy/strategy for these problems, most of them do not provide any guarantee for the performance of their resulting policy during the initial exploratory phase. This is a major obstacle in using learning algorithms in many different fields, such as online marketing, health sciences, finance, and robotics. Therefore, developing learning algorithms with *safety* guarantees can immensely increase the applicability of learning in solving decision problems. A policy generated by a learning algorithm is considered to be safe, if it is guaranteed to perform at least as well as a baseline. The baseline can be either a baseline value or the performance of a baseline strategy. It is important to note that since the policy is learned from data, it is a random variable, and thus, the safety guarantees are in high probability.

Safety can be studied in both *offline* and *online* scenarios. In the *offline* case, the algorithm learns the policy from a batch of data, usually generated by the current strategy or recent strategies of the company, and the question is whether the learned policy will perform as well as the current strategy or no worse than a baseline value, when it is deployed. This scenario has been recently studied heavily in both *model-based* (e.g., Petrik *et al.* [2016]) and *model-free* (e.g., Bottou *et al.* 2013; Thomas *et al.* 2015a,b; Swaminathan and Joachims 2015a,b) settings. In the model-based approach, we first use the batch of data and build a simulator that mimics the behavior of the dynamical system under study (hospital's ER, financial market, robot), and then use this simulator to generate data and learn the policy. The main challenge here is to have guarantees on the performance of the learned policy, given the error in the simulator. This line of research is closely related to the area of robust learning and control. In the model-free approach, we learn the policy directly from the batch of data, without building a simulator. This line of research is related to off-policy evaluation and control. While the

model-free approach is more suitable for problems in which we have access to a large batch of data, such as in online marketing, the model-based approach works better in problems in which data is harder to collect, but instead, we have good knowledge about the underlying dynamical system that allows us to build an accurate simulator.

In the *online* scenario, the algorithm learns a policy while interacting with the real system. Although (reasonable) online algorithms will eventually learn a good or an optimal policy, there is no guarantee for their performance along the way (the performance of their intermediate policies), especially at the very beginning, when they perform a large amount of *exploration*. Thus, in order to guarantee safety in online algorithms, it is important to control their exploration and make it more *conservative*. Consider a manager that allows our learning algorithm runs together with her company's current strategy (baseline policy), as long as it is safe, i.e., the loss incurred by letting a portion of the traffic handled by our algorithm (instead of by the baseline policy) does not exceed a certain threshold. Although we are confident that our algorithm will eventually perform at least as well as the baseline strategy, it should be able to remain alive (not terminated by the manager) long enough for this to happen. Therefore, we should make it more conservative (less exploratory) in a way not to violate the manager's safety constraint. This setting has been studied in the multi-armed bandit (MAB) [Wu *et al.*, 2016]. Wu *et al.* [2016] considered the baseline policy as a fixed arm in MAB, formulated safety using a constraint defined based on the performance of the baseline policy (mean of the baseline arm), and modified the UCB algorithm [Auer *et al.*, 2002] to satisfy this constraint.

In this paper, we study the notion of safety in *contextual linear bandits*, a setting that has application in many different fields including *personalized recommendation*. We first formulate safety in this setting, as a constraint that must hold *uniformly in time*, in Section 2. Our goal is to design learning algorithms that minimize regret under the constraint that at any given time, their expected sum of rewards should be above a fixed percentage of the expected sum of rewards of the baseline policy. This fixed percentage depends on the amount of risk that the manager is willing to take. In Section 3, we propose an algorithm, called *conservative linear UCB* (CLUCB), that satisfies the safety constraint. At each round, CLUCB plays the action suggested by the standard linear UCB (LUCB) algorithm (e.g., Dani *et al.* 2008; Rusmevichientong and Tsitsiklis 2010; Abbasi-Yadkori *et al.* 2011; Chu *et al.* 2011; Russo and Van Roy 2014), only if it satisfies the safety constraint for the worst choice of the parameter in the confidence set, and plays the action suggested by the baseline policy, otherwise. We prove an upper-bound for the regret of CLUCB, which can be decomposed into two terms. The first term is an upper-bound on the regret of LUCB that grows at the rate $\sqrt{T}\log(T)$. The second term is constant (does not grow with the horizon $T$) and accounts for the loss of being conservative in order to satisfy the safety constraint. This improves over the regret bound derived in Wu *et al.* [2016] for the MAB setting, where the regret of being conservative grows with time. In Section 4, we show how CLUCB can be extended to the case that the reward of the baseline policy is unknown without a change in its rate of regret. Finally in Section 5, we report experimental results that show CLUCB behaves as expected in practice and validate our theoretical analysis.

## 2 Problem Formulation

In this section, we first review the standard linear bandit setting and then introduce the conservative linear bandit formulation considered in this paper.

### 2.1 Linear Bandit

In the linear bandit setting, at any time $t$, the agent is given a set of (possibly) infinitely many actions/options $\mathcal{A}_t$, where each action $a \in \mathcal{A}_t$ is associated with a feature vector $\phi_a^t \in \mathbb{R}^d$. At each round $t$, the agent selects an action $a_t \in \mathcal{A}_t$ and observes a random reward $y_t$ generated as

$$y_t = \langle \theta^*, \phi_{a_t}^t \rangle + \eta_t, \tag{1}$$

where $\theta^* \in \mathbb{R}^d$ is the unknown reward parameter, $\langle \theta^*, \phi_{a_t}^t \rangle = r_{a_t}^t$ is the expected reward of action $a_t$ at time $t$, i.e., $r_{a_t}^t = \mathbb{E}[y_t]$, and $\eta_t$ is a random noise such that

**Assumption 1** *Each element $\eta_t$ of the noise sequence $\{\eta_t\}_{t=1}^{\infty}$ is conditionally $\sigma$-sub-Gaussian, i.e., $\mathbb{E}[e^{\zeta \eta_t} \mid a_{1:t}, \eta_{1:t-1}] \leq \exp(\zeta^2 \sigma^2 / 2), \ \forall \zeta \in \mathbb{R}$.*

The sub-Gaussian assumption implies that $\mathbb{E}[\eta_t \mid a_{1:t}, \eta_{1:t-1}] = 0$ and $\mathbf{Var}[\eta_t \mid a_{1:t}, \eta_{1:t-1}] \leq \sigma^2$.

Note that the above formulation contains time-varying action sets and time-dependent feature vectors for each action, and thus, includes the *linear contextual bandit* setting. In linear contextual bandit, if we denote by $x_t$, the state of the system at time $t$, the time-dependent feature vector $\phi_a^t$ for action $a$ will be equal to $\phi(x_t, a)$, the feature vector of state-action pair $(x_t, a)$.

We also make the following standard assumption on the unknown parameter $\theta^*$ and feature vectors:

**Assumption 2** *There exist constants $B, D \geq 0$ such that $\|\theta^*\|_2 \leq B$, $\|\phi_a^t\|_2 \leq D$, and $\langle \theta^*, \phi_a^t \rangle \in [0, 1]$, for all $t$ and all $a \in \mathcal{A}_t$.*

We define $\mathcal{B} = \left\{ \theta \in \mathbb{R}^d : \|\theta\|_2 \leq B \right\}$ and $\mathcal{F} = \left\{ \phi \in \mathbb{R}^d : \|\phi\|_2 \leq D, \ \langle \theta^*, \phi \rangle \in [0, 1] \right\}$ to be the parameter space and feature space, respectively.

Obviously, if the agent knows $\theta^*$, she will choose the optimal action $a_t^* = \arg\max_{a \in \mathcal{A}_t} \langle \theta^*, \phi_a^t \rangle$ at each round $t$. Since $\theta^*$ is unknown, the agent's goal is to maximize her cumulative expected rewards after $T$ rounds, i.e., $\sum_{t=1}^T \langle \theta^*, \phi_{a_t}^t \rangle$, or equivalently, to minimize its (pseudo)-regret, i.e.,

$$R_T = \sum_{t=1}^T \langle \theta^*, \phi_{a_t^*}^t \rangle - \sum_{t=1}^T \langle \theta^*, \phi_{a_t}^t \rangle, \tag{2}$$

which is the difference between the cumulative expected rewards of the optimal and agent's strategies.

## 2.2 Conservative Linear Bandit

The conservative linear bandit setting is exactly the same as the linear bandit, except that there exists a baseline policy $\pi_b$ (e.g., the company's current strategy) that at each round $t$, selects action $b_t \in \mathcal{A}_t$ and incurs the expected reward $r_{b_t}^t = \langle \theta^*, \phi_{b_t}^t \rangle$. We assume that the expected rewards of the actions taken by the baseline policy, $r_{b_t}^t$, are known (see Remark 1). We relax this assumption in Section 4 and extend our proposed algorithm to the case that the reward function of the baseline policy is not known in advance. Another difference between the conservative and standard linear bandit settings is the *performance constraint*, which is defined as follows:

**Definition 1 (Performance Constraint)** *At each round $t$, the difference between the performances of the baseline and the agent's policies should remain below a pre-defined fraction $\alpha \in (0, 1)$ of the baseline performance. This constraint may be written formally as*

$$\forall t \in \{1, \ldots, T\}, \quad \sum_{i=1}^t r_{b_i}^i - \sum_{i=1}^t r_{a_i}^i \leq \alpha \sum_{i=1}^t r_{b_i}^i \quad \text{or equivalently as} \quad \sum_{i=1}^t r_{a_i}^i \geq (1-\alpha) \sum_{i=1}^t r_{b_i}^i. \tag{3}$$

The parameter $\alpha$ controls the level of conservatism of the agent. Small values show that only small losses are tolerated and the agent should be overly conservative, whereas large values indicate that the manager is willing to take risk and the agent can be more explorative. Here, given the value of $\alpha$, the agent should select her actions in a way to both minimize her regret (2) and to satisfy the performance constraint (3). In the next section, we propose a linear bandit algorithm to achieve this goal with high probability.

**Remark 1.** Since the baseline policy is often our company's strategy, it is reasonable to assume that a large amount of data generated by this policy is available, and thus, we have an accurate estimate of its reward function. If in addition to this accurate estimate, we have access to the actual data, we can use them in our algorithms. The reason we do not use the data generated by the actions suggested by the baseline policy in constructing the confidence sets of our algorithm in Section 3 is mainly to keep the analysis simple. However, when dealing with the more general case of unknown baseline reward in Section 4, we construct the confidence sets using all available data, including those generated by the baseline policy. It is important to note that having a good estimate of the baseline reward function does not necessarily mean that we know the unknown parameter $\theta^*$. This is because the data used for this estimate has been generated by the baseline policy, and thus, may only provide a good estimate of $\theta^*$ in a limited subspace.

## 3 A Conservative Linear Bandit Algorithm

In this section, we propose a linear bandit algorithm, called *conservative linear upper confidence bound* (CLUCB), whose pseudocode is shown in Algorithm 1. CLUCB is based on the *optimism in the face of uncertainty* principle, and given the value of $\alpha$, minimizes the regret (2) and satisfies the performance constraint (3) with high probability. At each round $t$, CLUCB uses the previous

---

**Algorithm 1** CLUCB

---

**Input:** $\alpha, \mathcal{B}, \mathcal{F}$
**Initialize:** $S_0 = \emptyset$, $z_0 = \mathbf{0} \in \mathbb{R}^d$, and $\mathcal{C}_1 = \mathcal{B}$
**for** $t = 1, 2, 3, \cdots$ **do**
    Find $(a'_t, \widetilde{\theta}_t) \in \arg\max_{(a,\theta) \in \mathcal{A}_t \times \mathcal{C}_t} \ \langle \theta, \phi^t_a \rangle$
    Compute $L_t = \min_{\theta \in \mathcal{C}_t} \ \langle \theta, z_{t-1} + \phi^t_{a'_t} \rangle$
    **if** $L_t + \sum_{i \in S^c_{t-1}} r^i_{b_i} \geq (1-\alpha) \sum_{i=1}^t r^i_{b_i}$ **then**
        Play $a_t = a'_t$ and observe reward $y_t$ defined by (1)
        Set $z_t = z_{t-1} + \phi^t_{a_t}$, $S_t = S_{t-1} \cup t$, $S^c_t = S^c_{t-1}$
        Given $a_t$ and $y_t$, construct the confidence set $\mathcal{C}_{t+1}$ according to (5)
    **else**
        Play $a_t = b_t$ and observe reward $y_t$ defined by (1)
        Set $z_t = z_{t-1}$, $S_t = S_{t-1}$, $S^c_t = S^c_{t-1} \cup t$, $\mathcal{C}_{t+1} = \mathcal{C}_t$
    **end if**
**end for**

---

observations and builds a confidence set $\mathcal{C}_t$ that with high probability contains the unknown parameter $\theta^*$. It then selects the *optimistic action* $a'_t \in \arg\max_{a \in \mathcal{A}_t} \max_{\theta \in \mathcal{C}_t} \langle \theta, \phi^t_a \rangle$, which has the best performance among all the actions available in $\mathcal{A}_t$, within the confidence set $\mathcal{C}_t$. In order to make sure that the constraint (3) is satisfied, the algorithm plays the optimistic action $a'_t$, only if it satisfies the constraint for the worst choice of the parameter $\theta \in \mathcal{C}_t$. To make this more precise, let $S_{t-1}$ be the set of rounds $i < t$ at which CLUCB has played the optimistic action, i.e., $a_i = a'_i$. Similarly, $S^c_{t-1} = \{1, 2, \cdots, t-1\} - S_{t-1}$ is the set of rounds $j < t$ at which CLUCB has followed the baseline policy, i.e., $a_j = b_j$.

In order to guarantee that it does not violate constraint (3), at each round $t$, CLUCB plays the *optimistic action*, i.e., $a_t = a'_t$, only if

$$\min_{\theta \in \mathcal{C}_t} \left[ \sum_{i \in S^c_{t-1}} r^i_{b_i} + \left\langle \theta, \overbrace{\sum_{i \in S_{t-1}} \phi^i_{a_i}}^{z_{t-1}} \right\rangle + \langle \theta, \phi^t_{a'_t} \rangle \right] \geq (1-\alpha) \sum_{i=1}^t r^i_{b_i},$$

and plays the *conservative action*, i.e., $a_t = b_t$, otherwise. In the following, we describe how CLUCB constructs and updates its confidence sets $\mathcal{C}_t$.

### 3.1 Construction of Confidence Sets

CLUCB starts by the most general confidence set $\mathcal{C}_1 = \mathcal{B}$ and updates its confidence set only when it plays an optimistic action. This is mainly to simplify the analysis and is based on the idea that since the reward function of the baseline policy is known ahead of time, playing a baseline action does not provide any new information about the unknown parameter $\theta^*$. However, this can be easily changed to update the confidence set after each action. In fact, this is what we do in the algorithm proposed in Section 4. We follow the approach of Abbasi-Yadkori *et al.* [2011] to build confidence sets for $\theta^*$. Let $S_t = \{i_1, \ldots, i_{m_t}\}$ be the set of rounds up to and including round $t$ at which CLUCB has played the optimistic action. Note that we have defined $m_t = |S_t|$. For a fixed value of $\lambda > 0$, let

$$\widehat{\theta}_t = \left(\Phi_t \Phi_t^\mathsf{T} + \lambda I\right)^{-1} \Phi_t Y_t, \tag{4}$$

be the regularized least square estimate of $\theta$ at round $t$, where $\Phi_t = [\phi^{i_1}_{a_{i_1}}, \ldots, \phi^{i_{m_t}}_{a_{i_{m_t}}}]$ and $Y_t = [y_{i_1}, \ldots, y_{i_{m_t}}]^\top$. For a fixed confidence parameter $\delta \in (0,1)$, we construct the confidence set for the next round $t+1$ as

$$\mathcal{C}_{t+1} = \left\{ \theta \in \mathbb{R}^d : \|\theta - \widehat{\theta}_t\|_{V_t} \leq \beta_{t+1} \right\}, \tag{5}$$

where $\beta_{t+1} = \sigma \sqrt{d \log\left(\frac{1+(m_t+1)D^2/\lambda}{\delta}\right)} + \sqrt{\lambda}B$, $V_t = \lambda I + \Phi_t \Phi_t^\mathsf{T}$, and the weighted norm is defined as $\|x\|_V = \sqrt{x^\top V x}$ for any $x \in \mathbb{R}^d$ and any positive definite $V \in \mathbb{R}^{d \times d}$. Note that similar to the linear UCB algorithm (LUCB) in Abbasi-Yadkori *et al.* [2011], the sub-Gaussian parameter $\sigma$ and the regularization parameter $\lambda$ that appear in the definitions of $\beta_{t+1}$ and $V_t$ should also be given to the CLUCB algorithm as input. The following proposition (Theorem 2 in Abbasi-Yadkori *et al.* 2011) shows that the confidence sets constructed by (5) contain the true parameter $\theta^*$ with high probability.

**Proposition 1** *For the confidence set $\mathcal{C}_t$ defined by (5), we have $\mathbb{P}\big[\theta^* \in \mathcal{C}_t, \ \forall t \in \mathbb{N}\big] \geq 1 - \delta$.*

As mentioned before, CLUCB ensures that performance constraint (3) holds for all $\theta \in \mathcal{C}_t$ at all rounds $t$. As a result, if all the confidence sets hold (i.e., contain the true parameter $\theta^*$), CLUCB is guaranteed to satisfy performance constraint (3). Proposition 1 indicates that this happens with probability at least $1 - \delta$. It is worth noting that satisfying constraint (3) implies that CLUCB is at least as good as the baseline policy *at all rounds*. In this vein, Proposition 1 guarantees that, with probability at least $1 - \delta$, CLUCB performs no worse than the baseline policy *at all rounds*.

### 3.2 Regret Analysis of CLUCB

In this section, we prove a regret bound for the proposed CLUCB algorithm. Let $\Delta_{b_t}^t = r_{a_t^*}^t - r_{b_t}^t$ be the *baseline gap* at round $t$, i.e., the difference between the expected rewards of the optimal and baseline actions at round $t$. This quantity shows how sub-optimal the action suggested by the baseline policy is at round $t$. We make the following assumption on the performance of the baseline policy $\pi_b$.

**Assumption 3** *There exist $0 \leq \Delta_l \leq \Delta_h$ and $0 < r_l$ such that, at each round $t$,*

$$\Delta_l \leq \Delta_{b_t}^t \leq \Delta_h \qquad and \qquad r_l \leq r_{b_t}^t. \tag{6}$$

An obvious candidate for both $\Delta_h$ and $r_h$ is 1, as all the mean rewards are confined in $[0, 1]$. The reward lower-bound $r_l$ ensures that the baseline policy maintains a minimum level of performance at each round. Finally, $\Delta_l = 0$ is a reasonable candidate for the lower-bound of the baseline gap.

The following proposition shows that the regret of CLUCB can be decomposed into the regret of a linear UCB (LUCB) algorithm (e.g., Abbasi-Yadkori *et al.* 2011) and a regret caused by being conservative in order to satisfy the performance constraint (3).

**Proposition 2** *The regret of CLUCB can be decomposed into two terms as follows:*

$$R_T(CLUCB) \leq R_{S_T}(LUCB) + n_T \Delta_h, \tag{7}$$

*where $R_{S_T}(LUCB)$ is the cumulative (pseudo)-regret of LUCB at rounds $t \in S_T$ and $n_T = |S_T^c| = T - m_T$ is the number of rounds (in $T$ rounds) at which CLUCB has played a conservative action.*

**Proof:** From the definition of regret (2), we have

$$R_T(\text{CLUCB}) = \sum_{t=1}^{T} r_{a_t^*}^t - \sum_{t=1}^{T} r_{a_t}^t = \sum_{t \in S_T} (r_{a_t^*}^t - r_{a_t}^t) + \sum_{t \in S_T^c} \overbrace{(r_{a_t^*}^t - r_{b_t}^t)}^{\Delta_{b_t}^t} \leq \sum_{t \in S_T} (r_{a_t^*}^t - r_{a_t}^t) + n_T \Delta_h. \tag{8}$$

The result follows from the fact that for $t \in S_T$, CLUCB plays the exact same actions as LUCB, and thus, the first term in (8) represents LUCB's regret for these rounds. $\qquad\qquad\square$

The regret bound of LUCB for the confidence set (5) can be derived from the results of Abbasi-Yadkori *et al.* [2011]. Let $\mathcal{E}$ be the event that $\theta^* \in \mathcal{C}_t$, $\forall t \in \mathbb{N}$, which according to Proposition 1 holds w.p. at least $1 - \delta$. The following proposition provides a bound on $R_{S_T}(\text{LUCB})$. Since this proposition is a direct application of Thm. 3 in Abbasi-Yadkori *et al.* [2011], we omit its proof here.

**Proposition 3** *On event $\mathcal{E} = \{\theta^* \in \mathcal{C}_t, \ \forall t \in \mathbb{N}\}$, for any $T \in \mathbb{N}$, we have*

$$R_{S_T}(LUCB) \leq 4\sqrt{m_T d \log\left(\lambda + \frac{m_T D}{d}\right)} \times \left[B\sqrt{\lambda} + \sigma\sqrt{2\log(\frac{1}{\delta}) + d\log\left(1 + \frac{m_T D}{\lambda d}\right)}\right]$$

$$= O\left(d \log\left(\frac{D}{\lambda \delta} T\right)\sqrt{T}\right). \tag{9}$$

Now in order to bound the regret of CLUCB, we only need to find an upper-bound on $n_T$, i.e., the number of times that CLUCB deviates from LUCB and selects the action suggested by the baseline policy. We prove an upper-bound on $n_T$ in Theorem 4, which is the main technical result of this section. Due to space constraint, we only provide a proof sketch for Theorem 4 in the paper and report its detailed proof in Appendix A. The proof requires several technical lemmas that have been proved in Appendix C.

**Theorem 4** *Let $\lambda \geq \max(1, D^2)$. Then, on event $\mathcal{E}$, for any horizon $T \in \mathbb{N}$, we have*

$$n_T \leq 1 + 114d^2 \frac{(B\sqrt{\lambda}+\sigma)^2}{\alpha r_l(\Delta_l + \alpha r_l)} \left[\log\left(\frac{62d(B\sqrt{\lambda}+\sigma)}{\sqrt{\delta}(\Delta_l + \alpha r_l)}\right)\right]^2.$$

**Proof Sketch:** Let $\tau = \max\left\{1 \leq t \leq T \mid a_t \neq a'_t\right\}$ be the last round that CLUCB takes an action suggested by the baseline policy. We first show that at round $\tau$, the following holds:

$$\alpha \sum_{t=1}^{\tau} r_{b_t}^t \leq -(m_{\tau-1}+1)\Delta_l + 2\beta_\tau \left\|\phi_{a'_\tau}^\tau\right\|_{V_\tau^{-1}} + 2 \sum_{t \in S_{\tau-1}} \beta_t \left\|\phi_{a_t}^t\right\|_{V_t^{-1}} + 2\beta_\tau \left\|\phi_{a'_\tau}^\tau + \sum_{t \in S_{\tau-1}} \phi_{a_t}^t\right\|_{V_\tau^{-1}}.$$

Next, using Lemmas 7 and 8 (reported in Appendix C), and the Cauchy-Schwartz inequality, we deduce that

$$\alpha \sum_{t=1}^{\tau} r_{b_t}^t \leq -(m_{\tau-1}+1)\Delta_l + 8d(B\sqrt{\lambda}+\sigma)\log\left(\frac{2(m_{\tau-1}+1)}{\delta}\right)\sqrt{(m_{\tau-1}+1)}.$$

Since $r_{b_t}^t \geq r_l$ for all $t$, and $\tau = n_{\tau-1} + m_{\tau-1} + 1$, it follows that

$$\alpha r_l n_{\tau-1} \leq -(m_{\tau-1}+1)(\Delta_l + \alpha r_l) + 8d(B\sqrt{\lambda}+\sigma)\log\left(\frac{2(m_{\tau-1}+1)}{\delta}\right)\sqrt{(m_{\tau-1}+1)}. \qquad (10)$$

Note that $n_{\tau-1}$ and $m_{\tau-1}$ appear on the LHS and RHS of (10), respectively. The key point is that the RHS is positive only for a finite number of integers $m_{\tau-1}$, and thus, it has a finite upper bound. Using Lemma 9 (reported and proved in Appendix C), we prove that

$$\alpha r_l n_{\tau-1} \leq 114d^2 \frac{(B\sqrt{\lambda}+\sigma)^2}{\Delta_l + \alpha r_l} \times \left[\log\left(\frac{62d(B\sqrt{\lambda}+\sigma)}{\sqrt{\delta}(\Delta_l + \alpha r_l)}\right)\right]^2.$$

Finally, the fact that $n_T = n_\tau = n_{\tau-1} + 1$ completes the proof. $\qquad\square$

We now have all the necessary ingredients to derive a regret bound on the performance of the CLUCB algorithm. We report the regret bound of CLUCB in Theorem 5, whose proof is a direct consequence of the results of Propositions 2 and 3, and Theorem 4.

**Theorem 5** *Let $\lambda \geq \max(1, D^2)$. With probability at least $1 - \delta$, the CLUCB algorithm satisfies the performance constraint* (3) *for all $t \in \mathbb{N}$, and has the regret bound*

$$R_T(\text{CLUCB}) = O\left(d\log\left(\frac{DT}{\lambda\delta}\right)\sqrt{T} + \frac{K\Delta_h}{\alpha r_l}\right), \qquad (11)$$

*where $K$ is a constant that only depends on the parameters of the problem as*

$$K = 1 + 114d^2 \frac{(B\sqrt{\lambda}+\sigma)^2}{\Delta_l + \alpha r_l} \left[\log\left(\frac{62d(B\sqrt{\lambda}+\sigma)}{\sqrt{\delta}(\Delta_l + \alpha r_l)}\right)\right]^2.$$

**Remark 2.** The first term in the regret bound (11) is the regret of LUCB, which grows at the rate $\sqrt{T}\log(T)$. The second term accounts for the loss incurred by being conservative in order to satisfy the performance constraint (3). Our results indicate that this loss does not grow with time (since CLUCB acts conservatively only in a finite number of rounds). This is a clear improvement over the regret bound reported in Wu *et al.* [2016] for the MAB setting, in which the regret of being conservative grows with time. Furthermore, the regret bound of Theorem 5 clearly indicates that CLUCB's regret is larger for smaller values of $\alpha$. This perfectly matches the intuition that the agent must be more conservative, and thus, suffers higher regret for smaller values of $\alpha$. Theorem 5 also indicates that CLUCB's regret is smaller for smaller values of $\Delta_h$, because when the baseline policy $\pi_b$ is close to optimal, the algorithm does not lose much by being conservative.

---

**Algorithm 2** CLUCB2

---

  **Input:** $\alpha, r_l, \mathcal{B}, \mathcal{F}$
  **Initialize:** $n \leftarrow 0, \ z \leftarrow \mathbf{0}, w \leftarrow \mathbf{0}, v \leftarrow \mathbf{0}$ and $\mathcal{C}_1 \leftarrow \mathcal{B}$
  **for** $t = 1, 2, 3, \cdots$ **do**
    Let $b_t$ be the action suggested by $\pi_b$ at round $t$
    Find $(a'_t, \widetilde{\theta}) = \arg\max_{(a,\theta) \in \mathcal{A}_t \times \mathcal{C}_t} \langle \theta, \phi_a^t \rangle$
    Find $R_t = \max_{\theta \in \mathcal{C}_t} \langle \theta, v + \phi_{b_t}^t \rangle$ & $L_t = \min_{\theta \in \mathcal{C}_t} \langle \theta, z + \phi_{a'_t}^t \rangle + \alpha \max \left\{ \min_{\theta \in \mathcal{C}_t} \langle \theta, w \rangle, nr_l \right\}$
    **if** $L_t \geq (1 - \alpha)R_t$ **then**
      Play $a_t = a'_t$ and observe $y_t$ defined by (1)
      Set $\ z \leftarrow z + \phi_{a'_t}^t \ $ and $\ v \leftarrow v + \phi_{b_t}^t$
    **else**
      Play $a_t = b_t$ and observe $y_t$ defined by (1)
      Set $\ w = w + \phi_{b_t}^t \ $ and $\ n \leftarrow n + 1$
    **end if**
    Given $a_t$ and $y_t$, construct the confidence set $\mathcal{C}_{t+1}$ according to (15)
  **end for**

---

## 4 Unknown Baseline Reward

In this section, we consider the case where the expected rewards of the actions taken by the baseline policy, $r_{b_t}^t$, are *unknown* at the beginning. We show how the CLUCB algorithm presented in Section 3 should be changed to handle this case, and present a new algorithm, called CLUCB2. We prove a regret bound for CLUCB2, which is at the same rate as that for CLUCB. This shows that the lack of knowledge about the reward function of the baseline policy does not hurt our algorithm in terms of the rate of the regret. The pseudocode of CLUCB2 is shown in Algorithm 2. The main difference with CLUCB is in the condition that should be checked at each round $t$ to see whether we should play the optimistic action $a'_t$ or the conservative action $b_t$. This condition should be selected in a way that CLUCB2 satisfies constraint (3). We may rewrite (3) as

$$\sum_{i \in S_{t-1}} r_{a_i}^i + r_{a'_t}^t + \alpha \sum_{i \in S_{t-1}^c} r_{b_i}^i \geq (1 - \alpha)\big(r_{b_t}^t + \sum_{i \in S_{t-1}} r_{b_i}^i\big). \tag{12}$$

If we lower-bound the LHS and upper-bound the RHS of (12), we obtain

$$\min_{\theta \in \mathcal{C}_t} \langle \theta, \sum_{i \in S_{t-1}} \phi_{a_i}^i + \phi_{a'_t}^t \rangle + \alpha \min_{\theta \in \mathcal{C}_t} \langle \theta, \sum_{i \in S_{t-1}^c} \phi_{b_i}^i \rangle \geq (1 - \alpha) \max_{\theta \in \mathcal{C}_t} \langle \theta, \sum_{i \in S_{t-1}} \phi_{b_i}^i + \phi_{b_t}^t \rangle. \tag{13}$$

Since each confidence set $\mathcal{C}_t$ is built in a way to contain the true parameter $\theta^*$ with high probability, it is easy to see that (12) is satisfied whenever (13) is true.

CLUCB2 uses both optimistic and conservative actions, and their corresponding rewards in building its confidence sets. Specifically for any $t$, we let $\Phi_t = [\phi_{a_1}^1, \phi_{a_2}^2, \cdots, \phi_{a_t}^t]$, $Y_t = [y_1, y_2, \cdots, y_t]^\mathsf{T}$, $V_t = \lambda I + \Phi_t^\mathsf{T} \Phi_t$, and define the least-square estimate after round $t$ as

$$\widehat{\theta}_t = (\Phi_t \Phi_t^\mathsf{T} + \lambda I)^{-1} \Phi_t Y_t. \tag{14}$$

Given $V_t$ and $\widehat{\theta}_t$, the confidence set for round $t + 1$ is constructed as

$$\mathcal{C}_{t+1} = \left\{ \theta \in \mathcal{C}_t : \|\theta - \widehat{\theta}_t\|_{V_t} \leq \beta_{t+1} \right\}, \tag{15}$$

where $\mathcal{C}_1 = \mathcal{B}$ and $\beta_t = \sigma \sqrt{d \log\left(\frac{1+tD^2/\lambda}{\delta}\right)} + B\sqrt{\lambda}$. Similar to Proposition 1, we can easily prove that the confidence sets built by (15) contain the true parameter $\theta^*$ with high probability, i.e., $\mathbb{P}\big[\theta^* \in \mathcal{C}_t, \ \forall t \in \mathbb{N}\big] \geq 1 - \delta$.

**Remark 3.** Note that unlike the CLUCB algorithm, here we build nested confidence sets, i.e., $\cdots \subseteq \mathcal{C}_{t+1} \subseteq \mathcal{C}_t \subseteq \mathcal{C}_{t-1} \subseteq \cdots$, which is necessary for the proof of the algorithm. This can potentially increase the computational complexity of CLUCB2, but from a practical point of view, the confidence

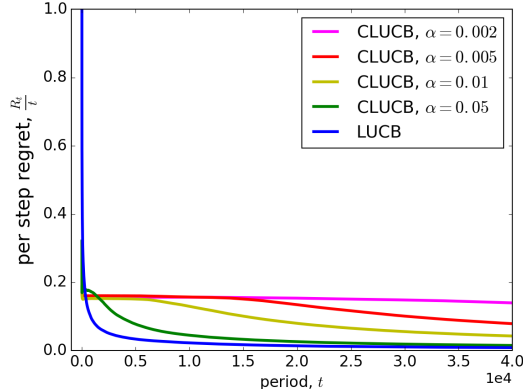

Figure 1: Average per-step regret (over $1,000$ runs) of LUCB and CLUCB for different values of $\alpha$.

sets become nested automatically after sufficient data has been observed. Therefore, the nested constraint in building the confidence sets can be relaxed after sufficiently large number of rounds.

The following theorem guarantees that CLUCB2 satisfies the safety constraint (3) with high probability, while its regret has the same rate as that of CLUCB and is worse than that of LUCB only up to an additive constant.

**Theorem 6** *Let $\lambda \geq \max(1, D^2)$ and $\delta \leq 2/e$. Then, with probability at least $1 - \delta$, CLUCB2 algorithm satisfies the performance constraint (3) for all $t \in \mathbb{N}$, and has the regret bound*

$$R_T(\textit{CLUCB2}) = O\left(d \log\left(\frac{DT}{\lambda\delta}\right)\sqrt{T} + \frac{K\Delta_h}{\alpha^2 r_l^2}\right), \tag{16}$$

*where $K$ is a constant that depends only on the parameters of the problem as*

$$K = 256 d^2 (B\sqrt{\lambda} + \sigma)^2 \left[\log\left(\frac{10 d (B\sqrt{\lambda} + \sigma)}{\alpha r_l(\delta)^{1/4}}\right)\right]^2 + 1.$$

We report the proof of Theorem 6 in Appendix B. The proof follows the same steps as that of Theorem 5, with additional non-trivial technicalities that have been highlighted there.

## 5   Simulation Results

In this section, we provide simulation results to illustrate the performance of the proposed CLUCB algorithm. We considered a time independent action set of 100 arms each having a time independent feature vector living in $\mathbb{R}^4$ space. These feature vectors and the parameter $\theta^*$ are randomly drawn from $\mathcal{N}(0, I_4)$ such that the mean reward associated to each arm is positive. The observation noise at each time step is also generated independently from $\mathcal{N}(0, 1)$, and the mean reward of the baseline policy at any time is taken to be the reward associated to the 10'th best action. We have taken $\lambda = 1, \delta = 0.001$ and the results are averaged over 1,000 realizations.

In Figure 1, we plot per-step regret (i.e., $\frac{R_t}{t}$) of LUCB and CLUCB for different values of $\alpha$ over a horizon $T = 40,000$. Figure 1 shows that per-step regret of CLUCB remains constant at the beginning (the conservative phase). This is because during this phase, CLUCB follows the baseline policy to make sure that the performance constraint (3) is satisfied. As expected, the length of the conservative phase decreases as $\alpha$ is increased, since the performance constraint is relaxed for larger values of $\alpha$, and hence, CLUCB starts playing optimistic actions more quickly. After this initial conservative phase, CLUCB has learned enough about the optimal action and its performance starts converging to that of LUCB. On the other hand, Figure 1 shows that per-step regret of CLUCB at the first few periods remains much lower than that of LUCB. This is because LUCB plays agnostic to the safety constraint, and thus, may select very poor actions in its initial exploration phase. In regard to this, Figure 2(a) plots the percentage of the rounds, in the first $1,000$ rounds, at which the safety constraint (3) is violated by LUCB and CLUCB for different values of $\alpha$. According to this figure,

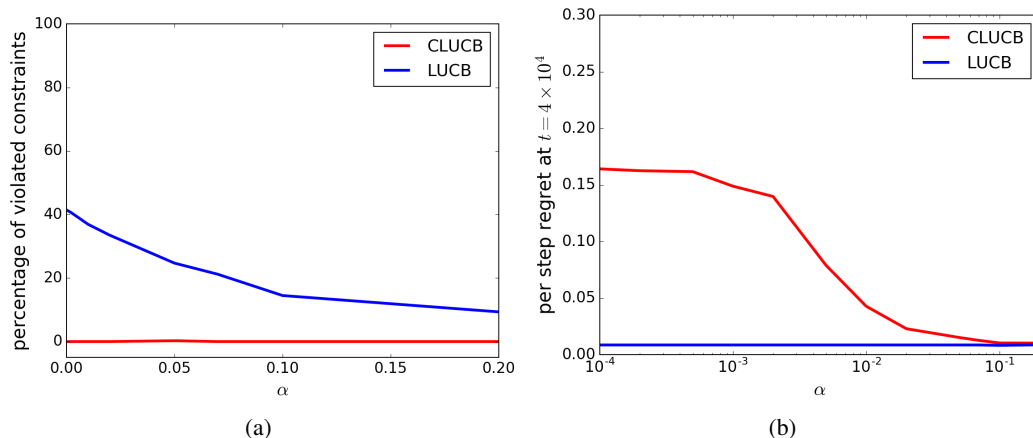

|     |     |
| :-: | :-: |
| (a) | (b) |

Figure 2: (a) Percentage of the rounds, in the first $1,000$ rounds, at which the safety constraint is violated by LUCB and CLUCB for different values of $\alpha$, (b) Per-step regret of LUCB and CLUCB for different values of $\alpha$, at round $t = 40,000$.

CLUCB satisfies the performance constraint for all values of $\alpha$, while LUCB fails in a significant number of rounds, specially for small values of $\alpha$ (i.e., tight constraint).

To better illustrate the effect of the performance constraint (3) on the regret of the algorithms, Figure 2(b) plots the per-step regret achieved by CLUCB at round $t = 40,000$ for different values of $\alpha$, as well as that for LUCB. As expected from our analysis and is shown in Figure 1, the performance of CLUCB converges to that of LUCB after an initial conservative phase. Figure 2(b) confirms that the convergence happens more quickly for larger values of $\alpha$, where the constraint is more relaxed.

## 6 Conclusions

In this paper, we studied the concept of *safety* in contextual linear bandits to address the challenges that arise in implementing such algorithms in practical situations such as personalized recommendation systems. Most of the existing linear bandit algorithms, such as LUCB [Abbasi-Yadkori *et al.*, 2011], suffer from a large regret at their initial exploratory rounds. This unsafe behavior is not acceptable in many practical situations, where having a reasonable performance at any time is necessary for a learning algorithm to be considered reliable and to remain in production.

To guarantee safe learning, we formulated a conservative linear bandit problem, where the performance of the learning algorithm (measured in terms of its cumulative rewards) at any time is constrained to be at least as good as a fraction of the performance of a baseline policy. We proposed a conservative version of LUCB algorithm, called CLUCB, to solve this constrained problem, and showed that it satisfies the safety constraint with high probability, while achieving a regret bound equivalent to that of LUCB up to an additive time-independent constant. We designed two versions of CLUCB that can be used depending on whether the reward function of the baseline policy is known or unknown, and showed that in each case, CLUCB acts conservatively (i.e., plays the action suggested by the baseline policy) only at a finite number of rounds, which depends on how suboptimal the baseline policy is. We reported simulation results that support our analysis and show the performance of the proposed CLUCB algorithm.

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
