[Supplementary Material]

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

# A   Proof of Theorem 4

**Theorem 4**   *Let $\lambda \geq \max(1, D^2)$. Then, on event $\mathcal{E}$, for any horizon $T \in \mathbb{N}$, we have*

$$n_T \leq 1 + 114d^2 \frac{(B\sqrt{\lambda} + \sigma)^2}{\alpha r_l(\Delta_l + \alpha r_l)} \left[ \log\left( \frac{62d(B\sqrt{\lambda} + \sigma)}{\sqrt{\delta}(\Delta_l + \alpha r_l)} \right) \right]^2.$$

**Proof:**   Let $\tau$ be the last round at which CLUCB plays conservatively (action suggested by the baseline policy), i.e., $\tau = \max\left\{1 \leq t \leq T \mid a_t \neq a_t'\right\}$. From Algorithm 1, at round $\tau$, we may write

$$\min_{\theta \in \mathcal{C}_\tau} \left\langle \theta, \phi_{a_\tau'}^\tau + \sum_{t \in S_{\tau-1}} \phi_{a_t}^t \right\rangle + \sum_{t \in S_{\tau-1}^c} r_{b_t}^t < (1-\alpha) \sum_{t=1}^\tau r_{b_t}^t.$$

or equivalently,

$$\alpha \sum_{t=1}^\tau r_{b_t}^t < \sum_{t \in S_\tau} r_{b_t}^t - \min_{\theta \in \mathcal{C}_\tau} \left\langle \theta, \phi_{a_\tau'}^\tau + \sum_{t \in S_{\tau-1}} \phi_{a_t}^t \right\rangle. \tag{17}$$

We may rewrite (17) as

$$\begin{aligned}
\alpha \sum_{t=1}^\tau r_{b_t}^t < &\sum_{t \in S_{\tau-1}} \left[ r_{b_t}^t - \langle \theta^*, \phi_{a_t}^t \rangle \right] + \left[ r_{b_\tau}^\tau - \langle \theta^*, \phi_{a_\tau'}^\tau \rangle \right] + \left\langle \theta^*, \phi_{a_\tau'}^\tau + \sum_{t \in S_{\tau-1}} \phi_{a_t}^t \right\rangle \\
&- \min_{\theta \in \mathcal{C}_\tau} \left\langle \theta, \phi_{a_\tau'}^\tau + \sum_{t \in S_{\tau-1}} \phi_{a_t}^t \right\rangle \\
= &\sum_{t \in S_{\tau-1}} \left[ r_{b_t}^t - \max_{\theta \in \mathcal{C}_t} \langle \theta, \phi_{a_t}^t \rangle + \max_{\theta \in \mathcal{C}_t} \langle \theta, \phi_{a_t}^t \rangle - \langle \theta^*, \phi_{a_t}^t \rangle \right] \\
&+ \left[ r_{b_\tau}^\tau - \max_{\theta \in \mathcal{C}_\tau} \langle \theta, \phi_{a_\tau'}^\tau \rangle + \max_{\theta \in \mathcal{C}_\tau} \langle \theta, \phi_{a_\tau'}^\tau \rangle - \langle \theta^*, \phi_{a_\tau'}^\tau \rangle \right] + \max_{\theta \in \mathcal{C}_\tau} \left\langle \theta^* - \theta, \phi_{a_\tau'}^\tau + \sum_{t \in S_{\tau-1}} \phi_{a_t}^t \right\rangle.
\end{aligned} \tag{18}$$

Note that for each $t \in S_{\tau-1}$, we have

$$r_{b_t}^t - \max_{\theta \in \mathcal{C}_t} \langle \theta, \phi_{a_t}^t \rangle \leq r_{b_t}^t - \langle \theta^*, \phi_{a_t^*}^t \rangle = -\Delta_{b_t}^t, \tag{19}$$

and similarly, for the round $\tau$, we have

$$r_{b_\tau}^\tau - \max_{\theta \in \mathcal{C}_\tau} \langle \theta, \phi_{a_\tau'}^\tau \rangle \leq r_{b_\tau}^\tau - \langle \theta^*, \phi_{a_\tau^*}^\tau \rangle = -\Delta_{b_\tau}^\tau. \tag{20}$$

Using inequalities (18) to (20), we may rewrite (17) as

$$\begin{aligned}
\alpha \sum_{t=1}^\tau r_{b_t}^t < &\sum_{t \in S_{\tau-1}} \left[ -\Delta_{b_t}^t + \max_{\theta \in \mathcal{C}_t} \langle \theta - \theta^*, \phi_{a_t}^t \rangle \right] - \Delta_{b_\tau}^\tau + \max_{\theta \in \mathcal{C}_\tau} \langle \theta - \theta^*, \phi_{a_\tau'}^\tau \rangle \\
&+ \max_{\theta \in \mathcal{C}_\tau} \left\langle \theta^* - \theta, \phi_{a_\tau'}^\tau + \sum_{t \in S_{\tau-1}} \phi_{a_t}^t \right\rangle \\
\leq &-(m_{\tau-1} + 1)\Delta_l + \max_{\theta \in \mathcal{C}_\tau} \langle \theta - \theta^*, \phi_{a_\tau'}^\tau \rangle + \sum_{t \in S_{\tau-1}} \max_{\theta \in \mathcal{C}_t} \langle \theta - \theta^*, \phi_{a_t}^t \rangle \\
&+ \max_{\theta \in \mathcal{C}_\tau} \left\langle \theta^* - \theta, \phi_{a_\tau'}^\tau + \sum_{t \in S_{\tau-1}} \phi_{a_t}^t \right\rangle \\
\leq &-(m_{\tau-1} + 1)\Delta_l + 2\beta_\tau \left\| \phi_{a_\tau'}^\tau \right\|_{V_\tau^{-1}} + 2 \sum_{t \in S_{\tau-1}} \beta_t \left\| \phi_{a_t}^t \right\|_{V_t^{-1}} + 2\beta_\tau \left\| \phi_{a_\tau'}^\tau + \sum_{t \in S_{\tau-1}} \phi_{a_t}^t \right\|_{V_\tau^{-1}} \\
\leq &-(m_{\tau-1} + 1)\Delta_l + 4\beta_\tau \left\| \phi_{a_\tau'}^\tau \right\|_{V_\tau^{-1}} + 2\beta_\tau \sum_{t \in S_{\tau-1}} \left( \left\| \phi_{a_t}^t \right\|_{V_t^{-1}} + \left\| \phi_{a_t}^t \right\|_{V_\tau^{-1}} \right)
\end{aligned} \tag{21}$$

$$\tag{22}$$

where the last inequality follows from triangle inequality and the fact that $\beta_\tau \geq \beta_t$ for all $t \leq \tau$. From Lemma 8 reported in Appendix C, it follows that $\left\|\phi_{a_t}^t\right\|_{V_t^{-1}} \geq \left\|\phi_{a_t}^t\right\|_{V_\tau^{-1}}$, for all $\in S_{\tau-1}$, and thus, (22) reduces to

$$\alpha \sum_{t=1}^\tau r_{b_t}^t < -(m_{\tau-1}+1)\Delta_l + 4\beta_\tau \left[ \left\|\phi_{a_\tau'}^\tau\right\|_{V_\tau^{-1}} + \sum_{t \in S_{\tau-1}} \left\|\phi_{a_t}^t\right\|_{V_t^{-1}} \right]. \tag{23}$$

In order to write the next equation more compactly, let us define $\Gamma$ as

$$\Gamma = \left[ \left\|\phi_{a_\tau'}^\tau\right\|_{V_\tau^{-1}}^2 + \sum_{t \in S_{\tau-1}} \left\|\phi_{a_t}^t\right\|_{V_t^{-1}}^2 \right].$$

From Cauchy-Schwarz inequality and Lemma 7, we have

$$\alpha \sum_{t=1}^\tau r_{b_t}^t \leq -(m_{\tau-1}+1)\Delta_l + 4\beta_\tau \sqrt{(m_{\tau-1}+1) \times \Gamma}$$

$$\leq -(m_{\tau-1}+1)\Delta_l + 4\beta_\tau \sqrt{2(m_{\tau-1}+1)d \log\left(1 + \frac{m_\tau D^2}{\lambda d}\right)}$$

$$= -(m_{\tau-1}+1)\Delta_l$$

$$+ 8\sqrt{(m_{\tau-1}+1)d \log\left(1 + \frac{(m_{\tau-1}+1)D^2}{\lambda d}\right)} \times \left[\sqrt{\lambda}B + \sigma\sqrt{d \log\left(\frac{\lambda + (m_{\tau-1}+1)D^2}{\lambda \delta}\right)}\right]$$

$$\leq -(m_{\tau-1}+1)\Delta_l + 8d(B\sqrt{\lambda} + \sigma) \log\left(\frac{2(m_{\tau-1}+1)}{\delta}\right) \sqrt{(m_{\tau-1}+1)},$$

where the last inequality follows from the fact that $\lambda \geq D^2$. On the other hand, since $r_{b_t}^t \geq r_l$ for all $t$, and $\tau = n_{\tau-1} + m_{\tau-1} + 1$, we may write

$$\alpha r_l n_{\tau-1} \leq -(m_{\tau-1}+1)(\Delta_l + \alpha r_l) + 8d(B\sqrt{\lambda} + \sigma) \log\left(\frac{2(m_{\tau-1}+1)}{\delta}\right) \sqrt{(m_{\tau-1}+1)}. \tag{24}$$

The RHS of (24) is only positive for a finite values of $m_\tau$, and thus, has a finite upper-bound. For $m = (m_{\tau-1}+1)$, $c_1 = 8d(B\sqrt{\lambda} + \sigma)$, $c_2 = \frac{2}{\delta}$, and $c_3 = (\Delta_l + \alpha r_l)$, Lemma 9 reported in Appendix C provides the following upper-bound on the RHS (and thus for the LHS) of (24):

$$\alpha r_l n_{\tau-1} \leq 114 d^2 \frac{(B\sqrt{\lambda} + \sigma)^2}{\Delta_l + \alpha r_l} \times \left[\log\left(\frac{62d(B\sqrt{\lambda} + \sigma)}{\sqrt{\delta}(\Delta_l + \alpha r_l)}\right)\right]^2.$$

The result follows from $n_T = n_\tau = n_{\tau-1} + 1$. □

# B  Proof of Theorem 6

**Theorem 6** *Assume that $\lambda \geq \max(1, D^2)$ and $\delta \leq 2/e$. With probability at least $1 - \delta$, the CLUCB2 algorithm satisfies the performance constraint (3) for all $t \in \mathbb{N}$, and has the following regret bound*

$$R_T(\text{CLUCB2}) = O\left(d\log\left(\frac{DT}{\lambda\delta}\right)\sqrt{T} + \frac{K\Delta_h}{\alpha^2 r_l^2}\right), \tag{25}$$

*where $K$ is a constant that depends only on the parameters of the problem as*

$$K = 256d^2(B\sqrt{\lambda} + \sigma)^2 \left[\log\left(\frac{10d(B\sqrt{\lambda} + \sigma)}{\alpha r_l(\delta)^{1/4}}\right)\right]^2 + 1.$$

**Proof:** Suppose the confidence sets do not fail, which is true with probability at least $1 - \delta$. Then, CLUCB2 satisfies constraints are all satisfied, since it ensures that those constraints are satisfied by the worst parameter in the confidence set at any time.

Similar to Proposition 2, we can decompose the regret of CLUCB2 as

$$R_T(\text{CLUCB2}) = \sum_{t \in S_T}\left(r_{a_t^*}^t - r_{a_t}^t\right) + \sum_{t \in S_T^c}\left(r_{a_t^*}^t - r_{b_t}^t\right) \leq \sum_{t \in S_T}\left(r_{a_t^*}^t - r_{a_t}^t\right) + n_T\Delta_h,$$

where $n_T = |S_T^c|$ is the number of times CLUCB2 follows the baseline policy in $T$ time steps. Now note that for $t \in S_T$, CLUCB2 is following the action suggested by LUCB and hence,

$$\sum_{t \in S_T}\left(r_{a_t^*}^t - r_{a_t}^t\right) \leq R_{S_T}(LUCB) \leq R_T(LUCB), \tag{26}$$

where $R_{S_T}(\text{LUCB})$ denotes the regret of LUCB played at time steps $t \in S_T$ which is upper bounded by the regret of LUCB played at all $T$ time steps. On the other hand, by Proposition 3, we have the following regret bound for LCUB:

$$R_T(\text{LUCB}) = O\left(d\log\left(\frac{D}{\lambda\delta}T\right)\sqrt{T}\right).$$

Thus, it follows that

$$R_T(\text{CLUCB2}) = O\left(d\log\left(\frac{D}{\lambda\delta}T\right)\sqrt{T} + n_T\Delta_h\right), \tag{27}$$

Note that according to (15), the confidence set $\mathcal{C}_t$ which CLUCB2 uses to find the optimistic action at round $t$ is built based not only on the observations made by previously played optimistic actions, but also by the observations made when the baseline policy has been followed at rounds before $t$. Therefore, the confidence set $\mathcal{C}_t$ used by CLUCB2 at round $t$ would be tighter than what LUCB would have had if it was applied only to rounds $i \in S_t$. Hence, the first inequality in (26) still holds.

Given (27), we only need to show that CLUCB2 follows the baseline policy only at a finite number of rounds. Let $\tau$ be the last round that CLUCB2 follows the baseline policy (plays conservatively), i.e., $\tau = \max\left\{1 \leq t \leq T \mid a_t \neq a_t'\right\}$. At time $\tau$, let

$$L_\tau = \min_{\theta \in \mathcal{C}_\tau}\left\langle\theta, \sum_{i \in S_{\tau-1}}\phi_{a_i}^i + \phi_{a_\tau'}^\tau\right\rangle + \alpha\max\left(\min_{\theta \in \mathcal{C}_\tau}\left\langle\theta, \sum_{i \in S_{\tau-1}^c}\phi_{b_i}^i\right\rangle, n_{\tau-1}r_l\right),$$

which satisfies $L_\tau \geq \min_{\theta \in \mathcal{C}_\tau}\left\langle\theta, \sum_{i \in S_{\tau-1}}\phi_{a_i}^i + \phi_{a_\tau'}^\tau\right\rangle + \alpha n_{\tau-1}r_l$, and

$$R_\tau = \max_{\theta \in \mathcal{C}_\tau}\left\langle\theta, \sum_{i \in S_{\tau-1}\cup\{\tau\}}\phi_{b_i}^i\right\rangle.$$

From Algorithm 2 at time $\tau$, we have $L_\tau < (1 - \alpha)R_\tau$ which with some simple algebra translates to

$$\alpha n_{\tau-1}r_l \leq \max_{\theta \in \mathcal{C}_\tau}\left\langle\theta, \sum_{i \in S_{\tau-1}\cup\{\tau\}}\phi_{b_i}^i\right\rangle - \min_{\theta \in \mathcal{C}_\tau}\left\langle\theta, \sum_{i \in S_{\tau-1}}\phi_{a_i}^i + \phi_{a_\tau'}^\tau\right\rangle - \alpha\max_{\theta \in \mathcal{C}_\tau}\left\langle\theta, \sum_{i \in S_{\tau-1}\cup\{\tau\}}\phi_{b_i}^i\right\rangle. \tag{28}$$

The rest of the proof is devoted to use (28) and prove a time independent upper bound on $n_{\tau-1}$. Unlike in the proof of Theorem 4, we rely on the nested property of the confidence sets built in (15) in this proof.

If the confidence sets do not fail (i.e., $\theta^* \in \mathcal{C}_\tau$), then since $\forall\, i \leq \tau : \mathcal{C}_\tau \subseteq \mathcal{C}_i$, we have

$$\max_{\theta \in \mathcal{C}_\tau} \Big\langle \theta, \sum_{i \in S_{\tau-1} \cup \{\tau\}} \phi_{b_i}^i \Big\rangle \leq \sum_{i \in S_{\tau-1} \cup \{\tau\}} \max_{\theta \in \mathcal{C}_\tau} \big\langle \theta, \phi_{b_i}^i \big\rangle \leq \sum_{i \in S_{\tau-1} \cup \{\tau\}} \max_{\theta \in \mathcal{C}_i} \big\langle \theta, \phi_{b_i}^i \big\rangle. \qquad (29)$$

First, it follows from (29) and the fact that $\big\langle \theta^*, \phi_{b_i}^i \big\rangle \geq r_l$ that

$$-\alpha \max_{\theta \in \mathcal{C}_\tau} \Big\langle \theta, \sum_{i \in S_{\tau-1} \cup \{\tau\}} \phi_{b_i}^i \Big\rangle \leq -\alpha(m_{\tau-1} + 1) r_l. \qquad (30)$$

On the other hand, by the definition of optimistic action $a_i'$ at round $i$ it follows that $\max_{\theta \in \mathcal{C}_i} \big\langle \theta, \phi_{b_i}^i \big\rangle \leq \max_{a \in \mathcal{A}_i} \max_{\theta \in \mathcal{C}_i} \big\langle \theta, \phi_a^i \big\rangle = \max_{\theta \in \mathcal{C}_i} \big\langle \theta, \phi_{a_i'}^i \big\rangle$. Then, from (29) it follows that

$$\max_{\theta \in \mathcal{C}_\tau} \Big\langle \theta, \sum_{i \in S_{\tau-1} \cup \{\tau\}} \phi_{b_i}^i \Big\rangle \leq \sum_{i \in S_{\tau-1}} \max_{\theta \in \mathcal{C}_i} \big\langle \theta, \phi_{a_i}^i \big\rangle + \max_{\theta \in \mathcal{C}_\tau} \big\langle \theta, \phi_{a_\tau'}^\tau \big\rangle. \qquad (31)$$

Furthermore, since $\forall\, i \leq \tau : \mathcal{C}_\tau \subseteq \mathcal{C}_i$, we have

$$\min_{\theta \in \mathcal{C}_\tau} \Big\langle \theta, \sum_{i \in S_{\tau-1}} \phi_{a_i}^i + \phi_{a_\tau'}^\tau \Big\rangle \geq \sum_{i \in S_{\tau-1}} \min_{\theta \in \mathcal{C}_\tau} \big\langle \theta, \phi_{a_i}^i \big\rangle + \min_{\theta \in \mathcal{C}_\tau} \big\langle \theta, \phi_{a_\tau'}^\tau \big\rangle$$

$$\geq \sum_{i \in S_{\tau-1}} \min_{\theta \in \mathcal{C}_i} \big\langle \theta, \phi_{a_i}^i \big\rangle + \min_{\theta \in \mathcal{C}_\tau} \big\langle \theta, \phi_{a_\tau'}^\tau \big\rangle. \qquad (32)$$

Combining (30), (31), (32) with (28) gives

$$\alpha n_{\tau-1} r_l \leq -\alpha(m_{\tau-1} + 1) r_l + \sum_{i \in S_{\tau-1}} \Big[ \max_{\theta \in \mathcal{C}_i} \big\langle \theta, \phi_{a_i}^i \big\rangle - \min_{\theta \in \mathcal{C}_i} \big\langle \theta, \phi_{a_i}^i \big\rangle \Big]$$

$$+ \Big[ \max_{\theta \in \mathcal{C}_\tau} \big\langle \theta, \phi_{a_\tau'}^\tau \big\rangle - \min_{\theta \in \mathcal{C}_\tau} \big\langle \theta, \phi_{a_\tau'}^\tau \big\rangle \Big]. \qquad (33)$$

Now, note that for any $i$, the reward of playing any action is in $[0, 1]$ and hence, $\big[ \max_{\theta \in \mathcal{C}_i} \big\langle \theta, \phi_{a_i}^i \big\rangle - \min_{\theta \in \mathcal{C}_i} \big\langle \theta, \phi_{a_i}^i \big\rangle \big] \leq 1$. On the other hand, since the confidence sets do not fail, we have

$$\max_{\theta \in \mathcal{C}_i} \big\langle \theta, \phi_{a_i}^i \big\rangle - \min_{\theta \in \mathcal{C}_i} \big\langle \theta, \phi_{a_i}^i \big\rangle = \max_{\theta \in \mathcal{C}_i} \big\langle \theta - \widehat{\theta}_{i-1}, \phi_{a_i}^i \big\rangle - \min_{\theta \in \mathcal{C}_i} \big\langle \widehat{\theta}_{i-1} - \theta, \phi_{a_i}^i \big\rangle$$

$$= 2 \max_{\theta \in \mathcal{C}_i} \big\langle \theta - \widehat{\theta}_{i-1}, \phi_{a_i}^i \big\rangle$$

$$\leq 2 \left\| \theta - \widehat{\theta}_{i-1} \right\|_{V_{i-1}} \left\| \phi_{a_i}^i \right\|_{V_{i-1}^{-1}}$$

$$\leq 2 \beta_i \left\| \phi_{a_i}^i \right\|_{V_{i-1}^{-1}}.$$

Now since $\beta_i$'s are non-decreasing, it follows that

$$\Big[ \max_{\theta \in \mathcal{C}_i} \big\langle \theta, \phi_{a_i}^i \big\rangle - \min_{\theta \in \mathcal{C}_i} \big\langle \theta, \phi_{a_i}^i \big\rangle \Big] \leq 2 \beta_\tau \left\| \phi_{a_i}^i \right\|_{V_{i-1}^{-1}}. \qquad (34)$$

Similarly, we can show that

$$\Big[ \max_{\theta \in \mathcal{C}_\tau} \big\langle \theta, \phi_{a_\tau'}^\tau \big\rangle - \min_{\theta \in \mathcal{C}_\tau} \big\langle \theta, \phi_{a_\tau'}^\tau \big\rangle \Big] \leq 2 \beta_\tau \left\| \phi_{a_\tau'}^\tau \right\|_{V_{\tau-1}^{-1}}. \qquad (35)$$

Substituting (34) and (35) in (33) gives

$$\alpha n_{\tau-1} r_l \leq -\alpha(m_{\tau-1} + 1) r_l + 2 \beta_\tau \left[ \sum_{i \in S_{\tau-1}} \left\| \phi_{a_i}^i \right\|_{V_{i-1}^{-1}} + \left\| \phi_{a_\tau'}^\tau \right\|_{V_{\tau-1}^{-1}} \right]. \qquad (36)$$

Bounding the RHS of (36) gives rise to another key difference between this proof and that of Theorem 4. Note that $V_i = \lambda I + \sum_{j \in S_i} \phi_{a_j}^j \big( \phi_{a_j}^j \big)^\top + \sum_{j \in S_i^c} \phi_{b_j}^j \big( \phi_{b_j}^j \big)^\top$ is built not only based on

the actions played at non-conservative rounds but also based on the actions played at conservative times ($j \in S_i^c$), and hence Lemma 7 cannot be directly used to bound the RHS of (36). Instead, we define for any $i$ : $\widetilde{V}_i = \lambda I + \sum_{j \in S_i} \phi_{a_j}^j \left(\phi_{a_j}^j\right)^\top$ which satisfies $V_i = \widetilde{V}_i + \sum_{j \in S_i^c} \phi_{b_j}^j \left(\phi_{b_j}^j\right)^\top$. According to Lemma 8, it follows that

$$\left\|\phi_{a_i}^i\right\|_{V_{i-1}^{-1}} \leq \left\|\phi_{a_i}^i\right\|_{\widetilde{V}_{i-1}^{-1}}, \quad \left\|\phi_{a'_\tau}^\tau\right\|_{V_{\tau-1}^{-1}} \leq \left\|\phi_{a'_\tau}^\tau\right\|_{\widetilde{V}_{\tau-1}^{-1}},$$

and hence, from (36) it follows that

$$\alpha n_{\tau-1} r_l \leq -\alpha(m_{\tau-1}+1)r_l + 2\beta_\tau \left[ \sum_{i \in S_{\tau-1}} \left\|\phi_{a_i}^i\right\|_{\widetilde{V}_{i-1}^{-1}} + \left\|\phi_{a'_\tau}^\tau\right\|_{\widetilde{V}_{\tau-1}^{-1}} \right]. \tag{37}$$

Note that $\tilde{V}_i$ is built based on the actions played at times in $S_i$. Now, similar to the proof of Theorem 4, we define

$$\Gamma = \left[ \sum_{i \in S_{\tau-1}} \left\|\phi_{a_i}^i\right\|_{\tilde{V}_{i-1}^{-1}}^2 + \left\|\phi_{a'_\tau}^\tau\right\|_{\tilde{V}_{\tau-1}^{-1}}^2 \right],$$

which by Lemma 7 satisfies

$$\Gamma \leq 2d \log \left( 1 + \frac{(m_{\tau-1}+1)D^2}{\lambda d} \right). \tag{38}$$

On the other hand, for $n_{\tau-1} \geq 3^1$ and $\delta \leq 2/e$, we have

$$\beta_\tau = \sigma \sqrt{d \log \left( \frac{1 + (m_{\tau-1}+1+n_{\tau-1})}{\delta} \right) D^2/\lambda} + B\sqrt{\lambda}$$

$$\leq (\sigma + B\sqrt{\lambda}) \sqrt{d \log \left( \frac{1 + (m_{\tau-1}+1)n_{\tau-1}D^2/\lambda}{\delta} \right)}, \tag{39}$$

where we used $\tau = m_{\tau-1} + n_{\tau-1} + 1$.

Using (38) and (39), and an application of Cauchy-Schwarz inequality on (37) gives

$$\alpha n_{\tau-1} r_l \leq -\alpha(m_{\tau-1}+1)r_l + 2\beta_\tau \sqrt{(m_{\tau-1}+1)\Gamma}$$

$$\leq -\alpha(m_{\tau-1}+1)r_l + 2\beta_\tau \sqrt{2d(m_{\tau-1}+1) \log \left( 1 + \frac{(m_{\tau-1}+1)D^2}{\lambda d} \right)}$$

$$\leq -\alpha(m_{\tau-1}+1)r_l + 3d(B\sqrt{\lambda}+\sigma) \log \left( \frac{2n_{\tau-1}}{\delta}(m_{\tau-1}+1) \right) \sqrt{(m_{\tau-1}+1)}, \tag{40}$$

where in the last inequality we used the fact that $\lambda \geq D^2$. Note that in contrary to the proof of Theorem 4 where only $m_{\tau-1}$ appeared on the RHS of (24), here both $n_{\tau-1}$ and $m_{\tau-1}$ both appear on the RHS. To bound $n_{\tau-1}$, we first provide an upper bound on the RHS of (40) in terms of $n_{\tau-1}$. For $m = (m_{\tau-1}+1)$, $c_1 = 3d(B\sqrt{\lambda}+\sigma)$, $c_2 = \frac{2n_{\tau-1}}{\delta}$ and $c_3 = \alpha r_l$, Lemma 9 provides the following upper bound on the RHS (and hence the LHS) of (40):

$$\alpha n_{\tau-1} r_l \leq 16d^2 \frac{(B\sqrt{\lambda}+\sigma)^2}{\alpha r_l} \left[ \log \left( \frac{24d(B\sqrt{\lambda}+\sigma)\sqrt{n_{\tau-1}}}{\sqrt{\delta}\alpha r_l} \right) \right]^2,$$

which is equivalent to

$$\sqrt{n_{\tau-1}} \leq 4d \frac{(B\sqrt{\lambda}+\sigma)}{\alpha r_l} \log \left( \frac{24d(B\sqrt{\lambda}+\sigma)\sqrt{n_{\tau-1}}}{\sqrt{\delta}\alpha r_l} \right). \tag{41}$$

Now, note that the LHS of (40) grows linearly with $\sqrt{n_{\tau-1}}$ while the RHS grows logarithmically. Thus, such an inequality holds only for a finite number of $n_{\tau-1}$'s. Lemma 10 applied with $x = \sqrt{n_{\tau-1}}$, $c_1 = 4d\frac{(B\sqrt{\lambda}+\sigma)}{\alpha r_l}$, and $c_2 = \frac{24d(B\sqrt{\lambda}+\sigma)}{\sqrt{\delta}\alpha r_l}$, gives the following upper bound on $n_{\tau-1}$:

$$n_{\tau-1} \le 256\frac{d^2(B\sqrt{\lambda}+\sigma)^2}{\alpha^2 r_l^2}\left[\log\left(\frac{10d(B\sqrt{\lambda}+\sigma)}{\alpha r_l(\delta)^{1/4}}\right)\right]^2.$$

Therefore, CLUCB2 follows the baseline policy only at

$$n_T = n_\tau = n_{\tau-1} + 1 \le 256\frac{d^2(B\sqrt{\lambda}+\sigma)^2}{\alpha^2 r_l^2}\left[\log\left(\frac{10d(B\sqrt{\lambda}+\sigma)}{\alpha r_l(\delta)^{1/4}}\right)\right]^2 + 1$$

rounds, and hence, according to (27), achieves a regret bound of

$$R_T(\text{CLUCB2}) = O\left(d\log\left(\frac{D}{\lambda\delta}T\right)\sqrt{T} + K\frac{\Delta_h}{\alpha^2 r_l^2}\right),$$

where

$$K = 256d^2(B\sqrt{\lambda}+\sigma)^2\left[\log\left(\frac{10d(B\sqrt{\lambda}+\sigma)}{\alpha r_l(\delta)^{1/4}}\right)\right]^2 + 1.$$

□

## C Technical Lemmas

In this Section, we report the technical lemmas used in the proofs of Theorems 4 and 6. The following lemma is used in the proof of Theorem 4.

**Lemma 7** *For given* $k \in \mathbb{N}, \lambda > 0$ *and any sequence* $X_1, X_2, \cdots, X_k$ *in* $\mathbb{R}^d$ *such that* $\forall\, i: \|X_i\|_2 \leq D$, *let* $V_0 = \lambda I$ *and* $V_i = \lambda I + \sum_{j=1}^{i} X_j X_j^\top$ *for* $1 \leq i \leq k$. *If* $\lambda \geq \max(1, D^2)$, *then we have*

$$\sum_{i=1}^{k} \|X_i\|_{V_{i-1}^{-1}}^2 \leq 2d \log\left(1 + \frac{kD^2}{\lambda d}\right). \tag{42}$$

**Proof:** This lemma is a direct application of Lemma 11 in Abbasi-Yadkori *et al.* [2011], and thus, we do not report its proof here. □

The following lemma is used in the proof of Theorem 4.

**Lemma 8** *Suppose* $\lambda > 0$ *and* $X_1, X_2, \cdots, X_m$ *are in* $\mathbb{R}^d$. *Define* $V_t = \lambda I + \sum_{i=1}^{t} X_i X_i^\intercal$ *for* $1 \leq t \leq n$. *Then, for any* $z\,in\mathbb{R}^d$ *and any* $t$, *we have*

$$\|z\|_{V_n^{-1}} \leq \|z\|_{V_t^{-1}}. \tag{43}$$

**Proof:** Note that we have

$$V_n = V_t + \sum_{i=t+1}^{n} X_i X_i^\intercal.$$

Now, let $U = [X_{t+1}, X_{t+2}, \cdots, X_n]$ so that $V_n = V_t + UU^\intercal$. By matrix inversion lemma, we have

$$V_n^{-1} = V_t^{-1} - V_t^{-1} U (I + U^\intercal U)^{-1} U^\intercal V_t^{-1}.$$

Thus, we have for any $z \in \mathbb{R}^d$

$$z^\intercal V_n^{-1} z = z^\intercal V_t^{-1} z - (U^\intercal V_t^{-1} z)^\intercal (I + U^\intercal U)^{-1} (U^\intercal V_t^{-1} z) \leq z^\intercal V_t^{-1} z,$$

where the last inequality follows from the fact that $(I + U^\intercal U)^{-1}$ is positive semidefinite. Thus, we have established that

$$\|z\|_{V_n^{-1}}^2 \leq \|z\|_{V_t^{-1}}^2.$$

□

In the proof of Theorems 4 and 6, we use the following lemma to bound the RHS of (23) and (40).

**Lemma 9** *For any* $m \geq 2$ *and* $c_1, c_2, c_3 > 0$, *the following holds*

$$-c_3 m + c_1 \sqrt{m} \log(c_2 m) \leq \frac{16 c_1^2}{9 c_3} \left[ \log\left(\frac{2 c_1 \sqrt{c_2} e}{c_3}\right) \right]^2. \tag{44}$$

**Proof:** Define the LHS of (44) as a function $g(m)$, $m \geq 2$, i.e.,

$$g(m) = -c_3 m + c_1 \sqrt{m} \log(c_2 m).$$

Firstly, note that we have

$$g'(m) = -c_3 + \frac{c_1 \left(2 + \log(c_2 m)\right)}{2 \sqrt{m}} \qquad \text{and} \qquad g''(m) = \frac{-c_1 \log(c_2 m)}{4 m \sqrt{m}}.$$

This implies that since $c_2 > 1$, $g$ is a differentiable concave function over its domain $[2, \infty)$, and thus, we can find $m^*$, the global maximum of function $g$. The first order condition implies that $g'(m^*) = 0$, which gives us

$$2 + \log(c_2 m^*) = \frac{2 c_3}{c_1} \sqrt{m^*}. \tag{45}$$

Plugging this into the definition of $g$, we obtain

$$g^* = \max_{m \geq 2} g(m) = g(m^*) = c_3 m^* - 2 c_1 \sqrt{m^*}.$$

Now, we use the change of variable $x = \frac{c_3}{2c_1}\sqrt{m^*}$, which by (C) gives us

$$g^* = \frac{4c_1^2}{c_3}(x^2 - x).\tag{46}$$

On the other hand, (45) becomes

$$2 + \log\left(\frac{4c_2 c_1^2}{c_3^2}\right) + 2\log(x) = 4x.$$

Taking $\exp$ from both sides gives us

$$\frac{e^{4x}}{x^2} = \frac{4c_1^2 c_2 e^2}{c_3^2}.$$

Now, since $x^2 \leq e^x$, for all $x \geq 0$, it follows from (C) that

$$\frac{4c_1^2 c_2 e^2}{c_3^2} = \frac{e^{4x}}{x^2} \geq \frac{e^{4x}}{e^x} = e^{3x},$$

which indicates that

$$x \leq \frac{1}{3}\log\left(\frac{4c_1^2 c_2 e^2}{c_3^2}\right).$$

Plugging this into (46) gives us

$$g^* \leq \frac{4c_1^2}{c_3}x^2 \leq \frac{4c_1^2}{9c_3}\left[\log\left(\frac{4c_1^2 c_2 e^2}{c_3^2}\right)\right]^2 = \frac{16c_1^2}{9c_3}\left[\log\left(\frac{2c_1\sqrt{c_2}e}{c_3}\right)\right]^2.$$

The statement follows from the fact that $g(m) \leq g^*$, for any $m \geq 2$. $\qquad\square$

Finally, the following Lemma is used in the proof of Theorem 6.

**Lemma 10** *Let $c_1$ and $c_2$ be two positive constants such that $\log(c_1 c_2) \geq 1$. Then, any $x > 0$ satisfying $x \leq c_1\log(c_2 x)$ also satisfies $x \leq 2c_1\log(c_1 c_2)$.*

**Proof:** Assume that $x \leq c_1\log(c_2 x)$ holds. Define $a = \frac{1}{c_1 c_2}$ and change the varible to $z = c_2 x$. Then, we have

$$az \leq \log(z).$$

Let $q(z) = az$ and $l(z) = \log(z)$ and define $z^* = 2/a\log(1/a)$. First, since $\log(t^2) \leq t$ for any $t > 0$, we have

$$\frac{1}{a} \geq \log\left(\frac{1}{a^2}\right) \Rightarrow \log\frac{1}{a} \geq \log\left(2\log\frac{1}{a}\right) \Rightarrow 2\log\frac{1}{a} \geq \log\left(\frac{2}{a}\log\frac{1}{a}\right).$$

By the definition of $z^*$, $q$ and $l$, the last inequality is equivalent to

$$q(z^*) \geq l(z^*).\tag{47}$$

Furthermore, since $\log(1/a) \geq 1$, for any $z \geq z^*$, we may write

$$q'(z) = a \geq \frac{a}{2\log(1/a)} = l'(z^*) \geq l'(z).\tag{48}$$

Thus, it follows from (47) and (48) that $q(z) \geq l(z)$ for all $z \geq z^*$. Thus, $az \leq \log(z)$ is possible only for $z \leq z^*$. Replacing the definition of $a$, $z$ and $z^*$, we deduce that $x \leq c_1\log(c_1 c_2)$ is possible only if $x \leq 2c_1\log(c_1 c_2)$. $\qquad\square$