[Reviews · NeurIPS 2017]

Reviewer 1



This paper studies a kind of contextual linear bandits with a conservative constraint, which enforces the player's cumulative reward at any time t to be at least (1-alpha)-times as larger as that of the given baseline policy for a give alpha. They consider two cases, the cases with known and unknown baseline rewards. For each case, they propose a UCB-type algorithm based on an existing algorithm for contextual linear bandits and prove its regret upper bound, which are composed of the regret caused by the base algorithm and the regret caused by satisfying the conservative constraint. They also conducted simulations of the algorithm with known baseline rewards, and checked that the algorithm really satisfies the conservative constraint. The conservative constraint, their algorithms and regret bounds seem natural. The graph of Figure 1(a) should be improved so as to be able to see the initial conservative phases of CLUCB more. The same experiments for CLUCB2 should be done because it works in a more realistic setting.

Reviewer 2



POST-REBUTTAL: The authors have answered my concerns and will clarify the point of confusion. I'm changing from a marginal accept to an accept. OLD REVIEW: Summary of the paper This paper proposes a "safe" algorithm for contextual linear bandits. This definition of safety assumes the existence of a current "baseline policy" for selecting actions. The algorithm is "safe" in that it guarantees that it will only select an action that differs from the action proposed by the baseline policy if the new action produces larger expected reward than the action proposed by the baseline policy. Due to the random nature of rewards, this guarantee is with high probability (probability at least 1-delta). Summary of review The paper is well written. It is an extremely easy read - I thank the authors for submitting a polished paper. The proposed problem setting and approach are novel to the best of my knowledge. The problem is well motivated and interesting. Sufficient theoretical and empirical justifications are provided to convince the reader of the viability of the proposed approach. However, I have some questions. I recommend at least weak acceptance, but would consider a stronger acceptance if I am misunderstanding some of these points. Questions 1. Definition 1 defines a desirable performance constraint. The high probability nature of this constraint should be clarified. Notice that line 135 doesn't mention that (3) must hold with high probability. This should be stated. 2. More importantly, the statement of *how* (3) must hold is critical, since right now it is ambiguous. During my first read it sounds like (3) must hold with probability 1-\delta. However this is *not* achieved by the algorithm. If I am understanding correctly (please correct me if I am wrong), the algorithm ensures that *at each time step* (3) holds with high probability. That is: \forall t \in \{1,\dotsc,T\}, \Pr \left ( \sum_{i=1}^t r_{b_i}^i - \sum_{i=1}^t r_{a_i}^t \leq \alpha \sum_{i=1}^t r_{b_i}^i \right ) NOT \Pr \left ( \forall t \in \{1,\dotsc,T\}, \sum_{i=1}^t r_{b_i}^i - \sum_{i=1}^t r_{a_i}^t \leq \alpha \sum_{i=1}^t r_{b_i}^i \right ) The current writing suggests the latter, which (I think) is not satisfied by the algorithm. In your response could you please clarify which of these you are claiming that your algorithm satisfies? 3. If your algorithm does in fact satisfy the former (the per-time step guarantee), then the motivation for the paper is undermined (this could be addressed by being upfront about the limitations of this approach in the introduction, without changing any content). Consider the actual guarantee that you provide in the domain used in the empirical study. You run the algorithm for 40,000 time steps with delta = 0.001. Your algorithm is meant to guarantee that "with high probability" it performs at least as well as the baseline. However, you only guarantee that the probability of playing a worse action will be at most 0.001 *at each time step*. So, you are guaranteeing that the probability of playing a worse action at some point will be at most 1-.999^40000 = (approximately) 1. That is, you are bounding the probability of an undesirable event to be at most 1, which is not very meaningful. This should be discussed. For now, I would appreciate your thoughts on why having a per-step high probability guarantee is important for systems where there are large numbers of time steps. If a single failure is damning then we should require a high probability guarantee that holds simultaneously for all time steps. If a single failure is not damning, but rather amortized cost over over thousands of time steps is what matters, then why are we trying to get per-time step high probability guarantees?

Reviewer 3



This paper presents a variant of linear UCB method for the contextual linear bandit problem, which is "conservative" in the sense that it will not violate the constraint that its performance is to be above a fixed percentage of a given baseline method. The authors prove some basic regret bounds on their proposed conservative method to establish their soundness. The idea of the method is straightforward, and the proofs are not particularly surprising, and the techniques used therein not overly innovative. Nonetheless, the problem formulation and proposed method are practical, and the results of the paper will likely benefit real world applications.